Using feces to indicate plastic pollution in terrestrial vertebrate species in western Thailand

Teampanpong Jiraporn 1 jiraporn.tea@ku.th
Duengkae Prateep 2
1 Department of Conservation, Faculty of Forestry, Kasetsart University , Chatuchak, Bangkok , Thailand
2 Department of Forest Biology, Faculty of Forestry, Kasetsart University , Chatuchak, Bangkok , Thailand
Oehlmann Jörg
Electronic publication date: 2024 Jun 25
Publication date: 2024
Volume: 12
Electronic Location ID: e17596
Received 2024 Jan 16; Accepted 2024 May 28
Copyright: © 2024 Teampanpong and Duengkae
Copyright year: 2024
Copyright holder: Teampanpong and Duengkae
License: This is an open access article distributed under the terms of the Creative Commons Attribution License, which permits unrestricted use, distribution, reproduction and adaptation in any medium and for any purpose provided that it is properly attributed. For attribution, the original author(s), title, publication source (PeerJ) and either DOI or URL of the article must be cited.
License URL: https://creativecommons.org/licenses/by/4.0/

Keywords: Feces, Microplastics, Terrestrial vertebrates, Western Thailand, Protected areas, Western Forest Complex, Plastic pollution

Funding: National Research Council of Thailand under Research and Innovation Grant 2020 This research was funded by the National Research Council of Thailand under Research and Innovation Grant 2020. The funders had no role in study design, data collection and analysis, decision to publish, or preparation of the manuscript.

==============================
Plastic pollution is a widespread and growing concern due to its transformation into microplastics (MPs), which can harm organisms and ecosystems. This study, aimed to identify plastic pollution in the feces of terrestrial vertebrates using convenience sampling both inside and outside protected areas in Western Thailand. We hypothesized that MPs are likely to be detectable in the feces of all vertebrate species, primarily in the form of small black fragments. We predicted varying quantities of MPs in the feces of the same species across different protected areas. Furthermore, we expected that factors indicating human presence, landscape characteristics, scat weight, and the MP abundance in water, soils, and sediments would influence the presence of plastics in feces. Among 12 terrestrial species studied, potential MPs were found in 41.11% of 90 samples, totaling 83 pieces across eight species including the Asian elephant (Elephas maximus), Eld’s deer (Rucervus eldii), Dhole (Cuon alpinus), Gaur (Bos gaurus), Sambar deer (Rusa unicolor), Wild boar (Sus scrofa), Northern red muntjac (Muntiacus vaginalis), and Butterfly lizard (Leiolepis belliana). Specifically, 3.61% of all potential MPs (three pieces) were macroplastics, and the remaining 96.39% were considered potential MPs with the abundance of 0.92 ± 1.89 items.scat−1 or 8.69 ± 32.56 items.100 g−1 dw. There was an association between the numbers of feces with and without potential plastics and species (χ2 = 20.88, p = 0.012). Most potential plastics were fibers (95.18%), predominantly black (56.63%) or blue (26.51%), with 74.70% smaller than two millimeters. Although there were no significant associations between species and plastic morphologies, colors, and sizes, the abundance classified by these characteristics varied significantly. FTIR identified 52.38% as natural fibers, 38.10% as synthetic fibers (rayon, polyurethane (PUR), polyethylene terephthalate (PET), polypropylene (PP), and PUR blended with cotton), and 9.52% as fragments of PET and Polyvinyl Chloride (PVC). Human-related factors were linked to the occurrence of potential plastics found in the feces of land-dwelling wildlife. This study enhances the understanding of plastic pollution in tropical protected areas, revealing the widespread of MPs even in small numbers from the areas distant from human settlements. Monitoring plastics in feces offers a non-invasive method for assessing plastic pollution in threatened species, as it allows for easy collection and taxonomic identification without harming live animals. However, stringent measures to assure the quality are necessitated to prevent exogenous MP contamination. These findings underscore the importance of raising awareness about plastic pollution in terrestrial ecosystems, especially regarding plastic products from clothing and plastic materials used in agriculture and irrigation systems.

Introduction

The invention of plastics has substantially benefited human society due to their stable chemical properties, good insulation, lightweight nature, and durability (Gu, Zhao & Johnson, 2020). Plastic production increased by 56% between 2000 and 2020, excluding synthetic fibers (Plastic Soup Foundation, 2022). Macroplastics and microplastics (MPs) are prevalent across all Earth’s ecosystems. MPs, defined as synthetic solid particles or polymeric matrices, range from 1 μm to 5 mm in size and are insoluble in water (Frias & Nash, 2019). They are classified as primary if they are manufactured to be small and as secondary if they result from the degradation of larger plastic items in the environment (Frias & Nash, 2019). MPs usually originate from the degradation of plastic debris, beginning with biofilm buildup by microorganisms, which greatly increases the weight of the plastic debris (Weinstein, Crocker & Gray, 2016). This is accompanied by a sharp decline in UV transmittance, leading to the emergence of MP fragments and fibers due to extensive cracking and pitting on the plastic surfaces (Weinstein, Crocker & Gray, 2016). Water (Julienne, Delorme & Lagarde, 2019) and beetle larvae (Gallitelli, Zauli & Scalic, 2022) were reported in causing MPs through plastic fragmentation (Julienne, Delorme & Lagarde, 2019).

Plastic pollution is an emerging environmental threat, jeopardizing ecosystem functions due to the widespread distribution of various organisms, thus posing risks to biodiversity and potentially causing unexpected ecological disruptions. These consequences can ultimately impact human health, society, the economy, and overall quality of life (Baho, Bundschuh & Futter, 2021). The ingestion of MPs, whether accidental or through trophic transfer in predator-prey relationships (Justino et al., 2023), poses risks to growth and reproduction (de Sá et al., 2018), potentially leading to reduced feeding capacity and energy reserves, and damage to the intestines of low trophic-level organisms (Wright, Thompson & Galloway, 2013). As recent research has indicated higher levels of MP contamination in terrestrial and freshwater ecosystems than in marine ecosystems (Horton et al., 2017), therefore studying MPs in terrestrial ecosystems can help answer questions about trophic transport, ingestion risks (Granek, Brander & Holland, 2020), and facilitate the design of management strategies to mitigate the sources of MPs for cleanup projects (Battisti et al., 2020).

Many studies have documented organisms eliminating MPs through feces. These studies included Harbor seals, Grey seals (Hudak & Sette, 2019), Beluga whales (Moore et al., 2020), fulmars (Provencher et al., 2018), sea turtles, pinnipeds (Meaza, Toyoda & Wise, 2021), coypu (Gallitelli et al., 2022), rabbits, mouflons (Álvarez-Méndez et al., 2024), European hedgehogs, wood mice, field voles, brown rats (Thrift et al., 2022), fishing cats (Ratnayaka et al., 2023), and humans (Schwabl et al., 2019). However, most of these studies have been conducted in temperate ecosystems, with only one study conducted in tropical Asia, highlighting the urgent research need due to increasing plastic pollution (Ratnayaka et al., 2023). With studies on MPs and their effects on terrestrial animals remain scarce (Gallitelli et al., 2022), especially in tropical zones such as Southeast Asia, evidence of MP excretion in feces could serve as another indicator to monitor MPs in terrestrial vertebrate species, where carcasses are rarely found and using invasive methods to capture them would not be justifiable.

Thailand ranked among the top five countries for per capita plastic waste generation in 2016 (Plastic Soup Foundation, 2022) and was identified as one of the 12 countries globally responsible for 52% of mismanaged plastic waste (Perreard et al., 2023). An estimated 70.1% of this waste ends up in freshwater and marine ecosystems, with tourism hotspots generating approximately 16,800 tons of mismanaged plastic waste annually (World Bank Group, 2021). As a result, Thailand faces a great risk of MP contamination in its terrestrial ecosystems. Despite this, research on MP contamination in terrestrial wildlife has been limited (Teampanpong & Duengkae, 2024). To address this gap, our study investigates the presence of potential MPs in the feces of terrestrial vertebrate species in Western Thailand, covering nine protected areas and outside protected areas. We quantified the excretion of potential MPs, classifying them by morphological characteristics (fiber, fragment, film, foam, pellet), colors, and sizes. The study examines environmental factors associated with the excretion of potential MPs. We hypothesize that potential MPs are detectable in the feces of all vertebrate species, predominantly as small black fragments. We expected varying quantities of MPs within the feces of the same species from different protected areas. Additionally, we anticipated that factors associated with human presence (proximities to landfills, tourist sites, local landmarks, villages, and unpaved roads), landscape characteristics (slope, elevation, land-use types, human population in the nearest village), weight of scat, and MP abundance in water, soils, and sediments, would influence the presence of plastics in feces. Our findings establish a baseline for monitoring plastic pollution in terrestrial vertebrate species through fecal analysis.

Materials and Methods

Study area

The study was conducted in Western Thailand, with the locations of feces collection illustrated in Fig. 1. The study area encompasses both protected and unprotected areas, including six national parks (Erawan, Khao Laem, Lam Klong Ngu, Sai Yok, Kheun Srinakarin, and Thong Pha Phum) and three wildlife sanctuaries (Salakpra, Huai Kha Khaeng, and Thung Yai West), accounting for approximately 30% of the Western Forest Complex. Thung Yai-Huai Kha Khaeng wildlife sanctuaries are the core areas of the Western Forest Complex, part of the first Natural World Heritage Site in Thailand. The dominant land uses in this region include dry dipterocarp forest, mixed deciduous forest, dry and evergreen forests, agricultural lands, and human settlements.

Figure 1 The study areas and locations of fecal collection.

Scope of the study areas including nine protected areas. Each data point indicate the locations where 90 fecal samples were collected from 12 vertebrate species in western Thailand. (The map was created in ArcMap 10.3. All point icons were derived from the software).

Field data collection

To conduct field collections of vertebrate scats, water, soils, and sediment in Thai protected areas, we obtained research permission from the Department of National Park, Wildlife, and Plant Conservation of Thailand (DNP: ID#0907.4/17863-26-Aug-2020). Samples were collected between August 2020 and June 2021. We followed the guidelines for feces, track, and sign identification (Thongnamchaima & Mather, 1997; World Wide Fund for Nature-Thailand, 2000) to ensure accurate species identification. For the fecal samples survey, a convenience sampling method was employed based on practical considerations of locations with easy access for quick data collection within limited time, vehicles, and manpower, involving park rangers and local people to identify locations where feces of terrestrial wildlife were most likely found. Only recent defecations were collected. A clean stainless steel spoon was used for subsampling feces to reduce soil contamination. The fecal samples were then placed in aluminum foil (Pérez-Guevara, Kutralam-Muniasamy & Shruti, 2021) and subsequently transferred into individual clear polyethylene ziplock bags. For reptiles and small- to medium-sized mammals (long-tailed macaques, wild boar, Burmese hare, sambar deer, Eld’s deer, Northern red muntjac, and dhole), the entire fresh dropping was collected. For large-sized mammals (Asian elephants, Banteng, Wild water buffalo, and Gaur), we sub-sampled fresh scats, targeting approximately 200 g of wet weight. During transportation to the laboratory at Kasetsart University’s Forest Biology Department in Bangkok, all samples were kept in an insulated ice cooler (~4 °C) to preserve them at the lowest possible temperature and were then transferred to a −20 °C freezer in the laboratory for further analysis.

Field collection for water, soil, and sediment at fecal sampling sites or nearby water bodies was conducted simultaneously after collecting fecal samples. Surface water (30 L) was collected using volume-reduced sampling (Hidalgo-Ruz et al., 2012) with an aluminum bucket and filtered through a 15 µm net. The water samples were then stored in clean glass bottles. For soil and sediment, soil cores of 5 cm in diameter and depth were taken from the surface using a composite sampling method (Scheurer & Bigalke, 2018), suitable for studying MPs in sediment (Zhang et al., 2020) and soil (Zhang & Liu, 2018). Three sub-samples of sediment within the 0–5 cm layer (Wang et al., 2020) along a stream bank or under shallow water, and three sub-samples of soil collected within the 0–10 cm from the surface (Möller, Löder & Laforsch, 2020) were combined into one composite sample per site (Zhang & Liu, 2018) to increase sampling heterogeneity (Möller, Löder & Laforsch, 2020). Stones larger than 1 cm were removed from the soil and sediment samples. Approximately 500 g of soils and sediments were collected and placed within a plastic bag, and transported to the laboratory. The stainless steel spoons used for sampling were rinsed with distilled water to prevent sample cross-contamination (Pérez-Guevara, Kutralam-Muniasamy & Shruti, 2021) for each of five batches and collected in glass bottles for testing the extraneous MPs remaining on the devices.

Sample preparation, classification and identification

Laboratory work commenced immediately after each field batch. The fecal samples were defrosted at room temperature in the collection bag, then air-dried inside the plastic bag to minimize contamination. They were subsequently heated at 50 °C for at least 3 days until reaching a stable dry weight, measured with 0.01 g accuracy. The entire pieces of dried scat from the Butterfly lizard and small to medium-sized mammals were used for analysis. For large mammals (Asian elephants, Banteng, Wild water buffalo, and Gaur), a subsample of the dried scat, up to 100 g, was used for analysis. The dried feces were soaked in distilled water before adding 30 ml of 4 M KOH, stirring, and incubating in the dark at room temperature in a sealed container for at least 24 h to pre-digest organic matter. KOH is recognized as suitable for the removal of animal tissues (Prata et al., 2019). Subsequently, 30% H2O2 was added and stirred for at least 24 h until complete digestion (Provencher et al., 2018), followed by the addition of saturated NaCl (30 ml) for density separation. The mixture was stirred and allowed to settle in the dark for another 24 h. The supernatant was vacuum-filtered, rinsed with distilled water through Whatman GF/C paper (1.2 µm pore size), labeled, and stored in a glass Petri dish to dry at 50 °C for 4 h and kept in aluminum foil for further analysis (Pérez-Guevara, Kutralam-Muniasamy & Shruti, 2021). The research protocol involving animal use was approved by the Institute Animal Care and Use Committee at Kasetsart University, approval number: ACKU63-ETC-001.

To examine plastics in water, the method described by Nuelle et al. (2014) was used. For soil and sediment, we utilized multiple protocols from Klein, Worch & Knepper (2015), Nuelle et al. (2014), Blair et al. (2019), and Qiu et al. (2016). All supernatants from water, soil, and sediment samples were vacuum-filtered through Whatman GF/C paper (1.2 µm pore size), labeled, stored in a glass Petri dish to dry at 50 °C for 4 h, before keeping them in aluminum foil for further analysis (Pérez-Guevara, Kutralam-Muniasamy & Shruti, 2021).

Samples were visually inspected under a ZEISS Stemi 508 stereomicroscope at 40× magnification. We identified and categorized pieces of plastic-like items based on their morphologies (fiber, fragment, film, and pellet), colors, and sizes. Size classes were adapted from Wang et al. (2017), including very small (≤0.05 mm), small (>0.05–0.5 mm), relatively small (>0.5–1 mm), moderate (>1–2 mm), relatively large (>2–3 mm), large (>3–4 mm), and very large (>4–5 mm). Potential MPs larger than 5 mm were categorized as macroplastics. The stereomicroscope could measure items as small as 0.01 mm. Due to limited funding, 21 out of 83 potential MPs (25.30%) were randomly selected for chemical property testing using Fourier Transform Infrared Spectrometry (FTIR: Type II; Perkin Elmer) with a range of 400–4,000 cm−1. Only readings with a confidence level of at least 60% were considered reliable (Lusher, McHugh & Thompson, 2013).

Quality assurance and quality control

To prevent procedural contamination of potential MPs during the fieldwork, we conducted MP tests on five distilled water samples for each of cleaning stainless steel spoons, cleaning soil cores, and plankton net. In the laboratory, all surfaces and equipment were thoroughly cleaned with distilled water, and all staff wore white cotton coats during operations. Moreover, laboratory blank controls for each of H2O2, distilled water, and air in the laboratory were conducted, once for each 10-laboratory batch. These blank samples underwent the same procedures as the collected samples. No plastic residues were found in the distilled water used to rinse field equipment. However, we found four microfibers in three out of 10 samples of distilled water in the laboratory (0.40 ± 0.66 items.sample−1) with blue, green, and red colors, and six microfibers in six out of 10 ambient laboratory air samples (0.60 ± 0.52 items.air sample−1) with light blue, blue, purple, black, and red colors. It is important to note that these air samples were collected under vacuum filtration for an hour (Wang et al., 2017), while our sample filtration under vacuum took less than 10 min per sample.

Data analysis

Due to limited fecal sample sizes and sampling methods, conducting extensive statistical tests was not fully feasible. Additionally, the application of FTIR analysis across the selected samples limited comparability. Therefore, the analysis focused primarily on visual data. R statistical software (version 4.2.3; R Core Team, 2023) was used for all statistical analyses. The presence of potential MPs in feces was reported as a percentage of occurrence (%). The abundance of potential MPs was calculated as items.scat−1, representing the total potential MPs per fecal sample, and items.100 g−1 dw, representing the total potential MPs per 100 grams of dry weight. Results were presented as mean ± SD. Pearson’s Chi-squared test with a simulated p-value was used to assess the associations between vertebrate species and the prevalence of potential MPs among different morphologies, colors, and sizes. The Kruskal-Wallis test was used for non-normally distributed data to differentiate the abundance of potential MPs in feces as classified by different morphologies, colors, and sizes. Two-way ANOVA with rank transformation was used to determine the effects of species and designated areas on the abundances of potential MPs, both per individual scats and per dry weight of scats.

Furthermore, the abundances of potential MPs in water, soils, and sediments, were calculated per samples and used as factors influencing the excretion of potential MPs in feces using a generalized linear model (GLM) with a negative binomial distribution to account for overdispersion with the R-package MASS (Venables & Ripley, 2002). The factors analyzed to investigate their influence on the occurrence of potential MPs in scats included species of terrestrial vertebrates, weight of scat, landscape characteristics (slope, elevation, land-use types, human population in the nearest village), factors indicating human presence (proximities to landfills, tourist sites, local landmarks, villages, and unpaved roads), and MP abundance in water, soils, and sediments at nearby locations to fecal collection sites.

Results

Presence, occurrence, and abundance of potential microplastics

In total, 90 scat samples were collected from 12 terrestrial vertebrate species, with eight species considered globally threatened (IUCN, 2023): six endangered (EN), two vulnerable (VU), and four least concern (LC). The species were: Asian elephant (Elephas maximus); Wild water buffalo (Bubalus arnee), Banteng (Bos javanicus); Eld’s deer (Rucervus eldii); Dhole (Cuon alpinus); Long-tailed macaque (Macaca fascicularis); Gaur (Bos gaurus); Sambar deer (Rusa unicolor); Wild boar (Sus scrofa); Burmese hare (Lepus peguensis); Northern red muntjac (Muntiacus vaginalis), and Butterfly lizard (Leiolepis belliana). Of these, 95.56% of samples were from inside protected areas and four (4.44%) from outside. One sample from outside protected areas was from a Long-tailed macaque found at a temple, and the remaining were from Asian elephants found along roads near protected areas and agricultural areas adjacent to protected areas.

Among the 90 fecal samples, potential plastics were not detected in the feces of Bantengs, Burmese hares, Wild water buffalo, and Long-tailed macaques. Only 41.11% of fecal samples of seven mammal species and one reptile species contained potential plastics, including Asian elephant, Eld’s deer, Dhole, Gaur, Sambar deer, Wild boar, Northern red muntjac, and Butterfly lizard (Fig. 2). The study found a significant association between fecal samples with and without potential plastics and species (χ2 = 20.88, p = 0.012). In total, 83 pieces of potential plastics were found in the feces, with an average of 0.92 ± 1.89 items.scat−1 or 8.69 ± 32.56 items.100 g−1 dw. Table 1 presents a summary by species of fecal samples, occurrences, and the abundance of potential plastics by individual samples and by dry weight of scat.

Figure 2 Comparison on the presence of potential plastics from feces of each wild terrestrial species.

Photo credit: Prateep Duengkae; Arnuparp Yhamdee.

Table 1 The summary categorizes fecal samples by species on occurrences of potential microplastics in these samples, and the abundance of potential microplastics, considering both the number of samples and their dry weight of scat.

Types of MPs/Species	Numbers of fecal samples (% total samples)	No. fecal samples with no potential MPs (% of samples per species)	No. fecal samples with potential MPs (% of MP occurrence in feces)	Total numbers of potential MPs	Abundance of potential MPs (item.scat−1)	Abundance of potential MPs (item.100 g−1 dw)	
Total	90 (100.00%)	53 (58.89%)	37 (41.11%)	83	0.92 ± 1.89	8.69 ± 32.56	
Banteng	4 (4.44%)	4 (100.00%)	0 (0.00%)	0	0.00 ± 0.00	0.00 ± 0.00	
Long-tailed macaque	1 (1.11%)	1 (100.00%)	0 (0.00%)	0	0.00	0.00	
Burmese hare	2 (2.22%)	2 (100.00%)	0 (0.00%)	0	0.00 ± 0.00	0.00 ± 0.00	
Wild water buffalo	1 (1.11%)	1 (100.00%)	0 (0.00%)	0	0.00	0.00	
Butterfly lizard	3 (3.33%)	1 (33.33%)	2 (66.67%)	4	1.33 ± 1.53	113.13 ± 142.18	
Dhole	2 (2.22%)	0 (0.00%)	2 (100.00%)	3	1.50 ± 0.71	55.96 ± 69.72	
Gaur	2 (2.22%)	0 (0.00%)	2 (100.00%)	7	3.50 ± 2.12	5.85 ± 3.37	
Sambar deer	18 (20.00%)	13 (72.22%)	5 (27.78%)	19	1.06 ± 3.51	7.68 ± 18.91	
Eld’s deer	9 (10.00%)	8 (88.89%)	1 (11.11%)	2	0.22 ± 0.67	4.73 ± 14.18	
Northern red muntjac	1 (1.11%)	0 (0.00%)	1 (100.00%)	3	3.00	34.88	
Wildboar	8 (8.89%)	5 (62.50%)	3 (37.50%)	5	0.62 ± 0.92	3.96 ± 7.60	
Asian elephant	39 (43.33%)	18 (46.15%)	21 (53.85%)	40	1.03 ± 1.22	1.82 ± 2.76	

Characteristics of potential plastics

In our findings, three macroplastics (2.41%) were identified, with sizes ranging from 5.14 to 5.45 mm (5.16 ± 0.09 mm), in the feces of the Asian elephant, equating to 0.02 ± 0.15 items.scat−1 or 0.06 ± 0.43 items.100 g−1 dw. The eighty potential MPs ranged from 0.14 to 4.65 mm, with an average length of 1.46 ± 1.01 mm. The majority of potential MPs (77.50%) were smaller than 2 mm. Moderately sized MPs (>1–2 mm) were the most common (36.25%), followed by relatively small (>0.5–1 mm; 26.25%) and small (0.05–0.5 mm; 15%). With the small numbers of macroplastics with sizes slightly larger MPs for few millimeters, we included them all into the analysis. The occurrence of potential MPs for each size and species was not significantly associated with fecal samples (χ2 = 41.33, p = 0.49). However, the abundances by individual scats differed significantly among size classes (H = 30.81, df = 6, p = 2.76 × 10−5), especially between the moderate and the large (>3–4 mm: p = 0.014), the very large (>4–5 mm, p = 0.005), and macroplastics (p = 0.005). Similarly, the abundances by scat dry weights varied significantly among different sizes (H = 29.41, df = 6, p = 5.10 × 10−5), notably between the moderate and the large (>3–4 mm, p = 0.02).

Fibers represented the majority of the plastic-like items, accounting for 79 out of 83 plastic items (95.18%). The remainder comprised three fragments (3.62%) found in the feces of the Asian elephant and one piece of film (1.20%) in the feces of a Sambar deer. The association between the occurrence of each morphology and species was not significant (χ2 = 6.68, p = 0.67). However, there were significant differences in abundance among morphologies, both per individual scats (H = 66.40, df = 2, p = 3.81 × 10−15), particularly between fibers and films (p = 8.4 × 10−10) and fibers and fragments (p = 7.5 × 10−9), and by dry weight of scat (H = 66.43, df = 2, p = 3.76 × 10−15), notably between fibers and films (p = 1.1 × 10−9) and fragments (p = 6.0 × 10−9).

Eight colors of potential MPs were identified in the feces, with black (56.63%) as the most prevalent color. Blue (26.51%) was the next most common. No significant association was found between the occurrence of potential MPs and colors across species (χ2 = 80.50, p = 0.06). However, there were significant differences in abundance among colors, both per individual scats (H = 118.87, df = 7, p < 2.2 × 10−16) and by dry weight (H = 117.7, df = 7, p < 2.2 × 10−16), especially between black and other colors (p < 0.01) except blue, and blue and transparent, light blue, yellow (p < 0.05), and green, red, white (p < 0.01). The quantities of potential MPs by species, morphologies, colors, and sizes are shown in Table 2 and Figs. 3 and 4 showed some examples of potential MPs from the study.

Table 2 The summary categorizes the total fecal samples, the feces with potential microplastics, the occurrences, and abundances by scat samples and by the dry weight of scat, classified according to species, morphologies, colors, and size classes.

Characteristics/Species	No. of fecal samples	No. feces with potential MPs occurrence by categories	Total potential MPs	Abundance of potential MPs (item.scat−1)	Abundance of potential MPs (item.100 g−1 dw)	
Microplastic morphologies					
Fiber		37	79	0.88 ± 1.80	8.60 ± 32.51	
Butterfly lizard	3	2 (66.67%)	4	1.33 ± 1.54	113.13 ± 142.18	
Dhole	2	2 (100.00%)	3	1.50 ± 0.71	55.97 ± 69.72	
Gaur	2	2 (100.00%)	7	3.50 ± 2.12	5.85 ± 3.37	
Sambar deer	18	5 (27.78%)	18	1.00 ± 3.27	7.47 ± 18.30	
Eld’s deer	9	1 (11.11%)	2	0.22 ± 0.67	4.73 ± 14.18	
Northern red muntjac	1	1 (11.11%)	3	3.00	34.88	
Wildboar	8	3 (37.50%)	5	0.63 ± 0.92	3.96 ± 7.60	
Asian elephant	39	19 (48.72%)	37	0.95 ± 1.23	1.73 ± 2.79	
Film		1	1	0.01 ± 0.11	0.04 ± 0.42	
Sambar deer	18	1 (5.56%)	1	0.06 ± 0.24	0.22 ± 0.93	
Fragment		3	3	0.03 ± 0.18	0.04 ± 0.22	
Asian elephant	39	3 (7.69%)	3	0.08 ± 0.27	0.09 ± 0.33	
Microplastic colors					
Black		29	47	0.52 ± 0.96	2.52 ± 8.78	
Dhole	2	2 (100.00%)	2	1.00 ± 0.00	29.65 ± 32.50	
Gaur	2	2 (100.00%)	4	2.00 ± 0.00	3.38 ± 0.13	
Sambar deer	18	3 (16.67%)	7	0.39 ± 1.20	4.50 ± 14.33	
Wildboar	8	2 (25.00%)	3	0.38 ± 0.74	3.35 ± 7.72	
Asian elephant	39	20 (51.28%)	31	0.80 ± 1.03	1.36 ± 2.04	
Blue	82	15	22	0.24 ± 0.66	5.57 ± 30.27	
Butterfly lizard	3	2 (66.67%)	4	1.33 ± 1.53	113.13 ± 142.18	
Dhole	2	1 (50.00%)	1	0.50 ± 0.71	26.32 ± 37.22	
Eld’s deer	9	1 (11.11%)	2	0.22 ± 0.67	4.73 ± 14.18	
Gaur	2	1 (50.00%)	1	0.50 ± 0.71	0.82 ± 1.16	
Sambar deer	18	2 (11.11%)	5	0.28 ± 0.96	1.45 ± 4.33	
Wildboar	8	1 (5.56%)	1	0.13 ± 0.35	0.30 ± 0.86	
Northern red muntjac	1	1 (100%)	2	2.00	23.30	
Asian elephant	39	6 (15.39%)	6	0.15 ± 0.37	0.33 ± 0.92	
Yellow		2	6	0.07 ± 0.47	0.21 ± 1.70	
Gaur	2	1 (50.00%)	2	1.00 ± 1.41	1.65 ± 2.33	
Sambar deer	18	1 (5.56%)	4	0.22 ± 0.94	0.88 ± 3.73	
Light blue		2	3	0.03 ± 0.23	0.10 ± 0.84	
Sambar deer	18	1 (5.56%)	2	0.11 ± 0.47	0.44 ± 1.86	
Asian elephant	39	1 (2.56%)	1	0.03 ± 0.16	0.03 ± 0.19	
Transparent		2	2	0.02 ± 0.15	0.11 ± 0.83	
Sambar deer	18	1 (5.56%)	1	0.06 ± 0.24	0.42 ± 1.79	
Wildboar	8	1 (12.50%)	1	0.13 ± 0.35	0.30 ± 0.86	
Red		1	1	0.01 ± 0.11	0.03 ± 0.29	
Asian elephant	39	1 (2.56%)	1	0.03 ± 0.16	0.07 ± 0.44	
Green		1	1	0.01 ± 0.11	0.01 ± 0.11	
Asian elephant	39	1 (2.56%)	1	0.03 ± 0.16	0.03 ± 0.16	
Cloudy white		1	1	0.01 ± 0.11	0.13 ± 1.23	
Northern red muntjac	1	1 (100.00%)	1	1.00	11.60	
Microplastic sizes					
Small (0.05–0.5 mm)	7	12	0.13 ± 0.56	0.99 ± 5.89	
Dhole	2	1 (50.00%)	1	0.50 ± 0.71	26.32 ± 37.22	
Asian elephant	39	3 (7.69%)	3	0.08 ± 0.27	0.12 ± 0.45	
Gaur	2	1 (50.00%)	3	1.50 ± 2.12	2.47 ± 3.49	
Sambar deer	18	1 (5.56%)	4	0.22 ± 0.94	0.88 ± 3.73	
Wildboar	8	1 (12.50%)	1	0.13 ± 0.35	1.37 ± 3.89	
Releatively small (>0.5–1 mm)		15	21	0.23 ± 0.58	1.70 ± 8.96	
Butterfly lizard	3	1 (33.33%)	1	0.33 ± 0.58	22.22 ± 38.49	
Dhole	2	1 (50.00%)	1	0.50 ± 0.71	26.32 ± 37.22	
Asian elephant	39	10 (25.64%)	14	0.36 ± 0.67	0.47 ± 0.97	
Gaur	2	2 (100.00%)	2	1.00 ± 0.00	1.69 ± 0.06	
Sambar deer	18	1 (5.56%)	3	0.17 ± 0.71	0.66 ± 2.79	
Medium (>1–2 mm)		19	29	0.32 ± 0.79	4.38 ± 29.09	
Butterfly lizard	3	1 (33.33%)	3	1.00 ± 1.73	90.91 ± 157.46	
Eld’s deer	9	1 (11.11%)	2	0.22 ± 0.67	4.73 ± 14.18	
Asian elephant	39	11 (28.21%)	14	0.36 ± 0.67	0.78 ± 2.10	
Gaur	2	1 (50.00%)	1	0.50 ± 0.71	0.82 ± 1.16	
Sambar deer	18	3 (16.67%)	7	0.39 ± 1.20	1.80 ± 5.11	
Northern red muntjac	1	1 (100.00%)	1	1.00	11.60	
Wildboar	8	1 (12.50%)	1	0.13 ± 0.35	0.30±0.86	
Relatively large (>2–3 mm)		11	11	0.12 ± 0.33	1.11 ± 6.39	
Dhole	2	1 (50.00%)	1	0.50 ± 0.71	3.33 ± 4.71	
Asian elephant	39	4 (10.26%)	4	0.10 ± 0.31	0.11 ± 0.32	
Sambar deer	18	3 (16.67%)	3	0.17 ± 0.38	3.91 ± 13.80	
Northern red muntjac	1	1 (100.00%)	1	1.00	11.60	
Wildboar	8	2 (25.00%)	2	0.25 ± 0.46	0.91 ± 1.81	
Large (>3–4 mm)		4	4	0.04 ± 0.21	0.22 ± 1.33	
Asian elephant	39	1 (2.56%)	1	0.03 ± 0.16	0.07 ± 0.44	
Gaur	2	1 (50.00%)	1	0.50 ± 0.71	0.87 ± 1.23	
Northern red muntjac	1	1 (100.00%)	1	1	11.6	
Sambar deer	18	1 (5.56%)	1	0.06 ± 0.24	0.22 ± 0.93	
Very large (>4–5 mm)		3	1	0.03 ± 0.18	0.18 ± 1.22	
Asian elephant	39	1 (2.56%)	1	0.03 ± 0.16	0.03 ± 0.16	
Sambar deer	18	1 (5.56%)	1	0.06 ± 0.24	0.22 ± 0.93	
Wildboar	8	1 (12.50%)	1	0.13 ± 0.35	1.37 ± 3.89	
Macroplastics (>5 mm)		3	3	0.03 ± 0.18	0.10 ± 0.56	
Asian elephant	39	3 (7.69%)	3	0.08 ± 0.27	0.24 ± 0.83	

Figure 3 The numbers of potential microplastics in feces classified by species in relation to morphologies (A), colors (B) and size classes (C).

Figure 4 (A–F) Photos of the most common charactericistics of potential microplastics detected in feces of wild terrestrial species in western Thailand.

From the 83 plastic items in fecal samples, we selectively chose 21 pieces (25.30%) for polymer testing based on morphologies and colors. The analysis revealed that 52.38% of the tested potential MPs were fibers, including natural fibers (cotton, ramie, flax, agave), and 38.10% were synthetic fibers (rayon, polyurethane: PUR, polyethylene terephthalate: PET, polypropylene: PP, and PUR blended with cotton). The remaining 9.52% were fragments of PET and polyvinyl chloride (PVC). Figure 5 displays some results from the FTIR analysis of PET, PP, and cotton. Among the plastics identified by FTIR, PET (33.33%) was the most abundant, followed by PUR and PP (22.22% each), and PU and PVC (11.11% each).

Figure 5 FT-IR analysis and photos of the natural fibers and microplastics detected in fecal samples.

Factors related to occurrence of potential microplastics in feces

The majority of fecal samples were collected from Salakpra, followed by Erawan, Sai Yok, Kheun Srinakarin, Huai Kha Khaeng, outside protected areas, Thung Yai West, and Thong Pha Phum. Potential MPs were present in 100% of fecal samples from Thong Pha Phum and Thung Yai West (Fig. 6). There was no statistically significant association between the designated areas and the presence of plastic-like items (χ2 = 11.00, p = 0.13). However, there was a significant association between species and designated areas for all fecal samples with plastic presence (χ2 = 207.28, p = 5.0 × 10−3). Furthermore, the abundance of potential MPs per scat was significantly affected by both vertebrate species (F11 = 2.10, p = 0.03) and designated areas (F7 = 2.48, p = 0.03), as well as by the dry weight of scat for both vertebrate species (F11, 2.64, p = 0.007) and designated areas (F7, 2.50, p = 0.02). Figure 7 shows the abundances of potential MPs compared by species and protected areas.

Figure 6 (A–D) The numbers of fecal sample, the number of feces with potential microplastics, the percent of feces with and without potential microplastics, and the abundance of potential microplastics in feces of all species classified by protected areas.

PA, Protected areas; NP, National Park; WS, Wildlife Sanctuary.

Figure 7 The abundance of potential microplastics as classified by species and protected areas by individual scat (A) and by dry weight of scat (B).

PA, Protected areas; NP, National Park; WS, Wildlife Sanctuary.

Using a generalized linear model with a negative binomial distribution, the occurrence levels of potential MPs in each sample were found to increase with the MP abundance in soil and the human population in nearby villages (Table 3). It is noted that MPs in water, soils, and sediments were not found in 4.44%, 5.56%, and 5.56% of the total samples, respectively, with the average abundance of 18.74 ± 19.86, 6.28 ± 3.91, and 7.00 ± 6.62 items per sample, respectively.

Table 3 Factors affecting the occurrence of MPs in feces analyzed by Generalized linear model (GLM: negative binomial).

Parameter	Estimate	SE	Z	P	
Intercept	−1.15	0.45	−3.36	7.77 × 10-4	
MP abundance in soil	0.09	0.04	2.17	0.03	
Number of population	0.002	0.0008	2.24	0.03	
Note:

Null deviance = 93.58 at df = 89, Residual deviance = 80.08 at df = 87 AIC = 229.91, Theta = 0.77, SE = 0.29, log-likelihood = −221.91.

Discussion

Our study provides the first evidence of plastic excretion in the feces of terrestrial vertebrate species in Western Thailand. While potential plastics were not found in the feces of Banteng, Burmese hare, Wild water buffalo, and Long-tailed macaque, 83 potential MPs were recorded in 41.11% of the 82 fecal samples across eight terrestrial vertebrate species. Given the unequal number of fecal samples among species, direct comparisons may not be suitable; however, our results suggest that carnivorous species (Dhole and Butterfly lizard) may have a higher concentration of potential MPs in their feces than herbivores (Asian elephants, Gaur, Eld’s deer, Sambar deer, Northern red muntjac) and the omnivore (Wild boar). This indicates that the ingestion of plastics is not confined to any specific dietary preference (Thrift et al., 2022).

The low excretion of potential MPs in the feces of Asian elephants, despite their large body size and potential for greater exposure to plastics in the human-dominated landscapes of western Thailand, could be due to the ability of larger mammals to defecate potential MPs along with other inedible and indigestible items (Lusher et al., 2018). Although Asian elephants are known to ingest large pieces of plastics and excrete them in their feces (Katlam et al., 2022), and large mammals are believed to retain plastics longer in their digestive systems (Carlsson, Singdahl-Larsen & Lusher, 2021), our study was not designed to verify that large pieces of plastics are the main source of MPs in the feces of Asian elephants. Further necropsy investigations would be needed to confirm this (Carlsson, Singdahl-Larsen & Lusher, 2021).

While the composition of feces varies among different taxonomic groups and habitats (Thrift et al., 2022), we compared the abundance of potential MPs in terrestrial wildlife in western Thailand (0.92 ± 1.89 items.scat−1 or 8.69 ± 32.56 items.100 g−1 dw) to that reported in other animals. For instance, MP excretion in the feces of grey seal (Halichoerus grypus), monk seal (Monachus monachus), whale shark (Rhincodon typus), and three rabbit species (desert cottontail Sylvilagus audubonii, brush rabbit S. bachmani, and black-tailed jackrabbit Lepus californicus) showed reported values of 1.08 ± 1.01 items.scat−1 (Desclos-Dukes, Gutterworth & Cogan, 2022), 13.83 items.scat−1 (Hernandez-Milian et al., 2023), 3.20 ± 3.6 items.scat−1 or 1.11 ± 0.58 items.g−1 (Yong et al., 2021), and 5.6 ± 6.1 items.g−1 (Alvarez-Andrade et al., 2023), respectively.

The detection of MPs in scats confirms that terrestrial vertebrate species can excrete such contaminants (Bessa et al., 2019). We believe only a low proportion of ingested MPs is excreted, as illustrated by Moore et al. (2020), who found MPs in the gastrointestinal tract of seven Beluga whales (97 ± 42 items.scat−1), of these, only two whales had feces with micropalstic contamination, one containing two MPs and the other none, despite having 147 and 85 MPs in their gastrointestinal tract, respectively. This reflects a notably low proportion of plastic ingestion to excretion.

Morphologies, colors, sizes, and polymer types

In our study, fibers were the most abundant type of MPs, predominantly in black and blue. Most MPs were smaller than 2 mm. The majority were natural fibers (cotton, ramie, flax, agave) and synthetic fibers (rayon, PUR, PET, PP, PUR blended with cotton), with only a few fragmented PET and PVC pieces. Fibers were also the predominant type of MPs in fecal samples of various animals, including migratory birds (80%; Masiá, Ardura & Garcia-Vazquez, 2019), Northern fulmar (75%; Provencher et al., 2018), Gentoo penguin (58%; Bessa et al., 2019), grey seal (76.5%; Nelms et al., 2019), three rabbit species (72%; Alvarez-Andrade et al., 2023), and coyote (Canis latrans) and Lesser anteater (Tamandua mexicana) for 100% (Mendoza-Arroyo et al., 2024). This contrasts with the findings of Yang et al. (2019) and Donohue et al. (2019), where 96.65% and 68.15% of the MPs were fragments in the feces of whale sharks in the Philippines and fur seals in the USA, respectively, with fibers accounting for only 3.35% and 31.85%.

Black was the more prevalent color in our fecal samples, aligning with the findings of Desclos-Dukes, Gutterworth & Cogan (2022), Le Guen et al. (2020), and Alvarez-Andrade et al. (2023), who reported microfibers in grey seal scats at 42% black, in king penguins at 50% black (18% blue), and in three rabbit species at 24% black (22% blue), respectively. However, several studies have reported a higher proportion of blue MPs, including in migratory birds (Masiá, Ardura & Garcia-Vazquez, 2019), Northern fulmar (Provencher et al., 2018), and grey seal (Nelms et al., 2019), with blue-to-black ratios of 40:10, 41.57:0, and 53:17, respectively.

MPs smaller than 2 mm were the most frequently found in the scats of vertebrate species in our study, similar to findings in Northern fur seals (Callorhinus ursinus; 89.70%; Donohue et al., 2019), Gentoo penguins (84.21%; Bessa et al., 2019), whale sharks (91.36%; Yang et al., 2019), and three species of rabbits (91%; Alvarez-Andrade et al., 2023). The predominance of small MPs (≤2 mm) might be attributed to their increased bioavailability, facilitating trophic transfer of this pollution from lower trophic level organisms (Wright, Thompson & Galloway, 2013). Unfortunately, without a dietary analysis of the collected scats from the eight vertebrate species in our study, it cannot be definitively stated that the MPs in scats were derived from both intentional and unintentional feeding, as well as trophic transfer from contaminated food sources lower in the food chain.

Our study found 52.38% natural fibers (cotton, ramie, flax, agave), 38.10% synthetic fibers (rayon, PUR, PET, PP, and PUR blended with cotton), and 9.52% fragments of PET and PVC. PET fibers, which account for over 70% of global fiber production (Periyasamy & Tehrani-Bagha, 2022), are more commonly found in the environment than PP (Carney Almroth et al., 2018). Fiber fragments, both natural and synthetic, often derive from garments and home textiles during washing, drying, and wearing, and are becoming a significant source of plastic pollution, exacerbated by fast fashion and population growth (Periyasamy & Tehrani-Bagha, 2022; Thrift et al., 2022). It has been reported that washed synthetic fabrics can release between 124 and 308 mg of microfibers for every kilogram of clothing (De Falco et al., 2018) and up to 110,000 pieces of PET per garment per wash from fleece (Carney Almroth et al., 2018). Synthetic PUR fibers, known for their excellent elasticity and smooth draping, are commonly used in textiles and marketed under the Lycra® brand (De Oliveira et al., 2023). PP is favored in the market for its durability and affordability (Periyasamy, Viková & Vik, 2020) and is widely used in the manufacture of furniture such as carpets, sofas, and chairs (Dris et al., 2017), as well as in ropes, nets, fishing gear, and packaging and labeling for food (Zhang et al., 2021). PVC and PUR have been reported to significantly inhibit seed germination and seedling growth in Nelumbo nucifera more than PET during 7-day toxicity tests (Esterhuizen & Kim, 2022). Further research is necessary to explore the hazards of these polymers on terrestrial vertebrate species.

Factors influencing excretion of potential microplastics in vertebrate feces

The occurrence of potential plastics in the feces of terrestrial vertebrate species was influenced by the MP abundance in soils and the human population size of the nearest village. These findings suggest that MPs in vertebrate feces are primarily originating from MP contaminated in the environment (in this case from soil), particularly near human-dominated landscapes. Here, MPs can be released from activities such as clothes washing (Yang et al., 2019) and from the degradation of plastic litter and debris by water (Julienne, Delorme & Lagarde, 2019) and beetle larvae (Gallitelli, Zauli & Scalic, 2022). Open dumps were identified as the main sources of MPs in the Mae Klong watershed, Thailand (World Bank Group, 2021). Higher micropalstic concentrations in soils near villages with large human populations, especially in agricultural areas, might result from plastic-derived human activities, including wastewater discharge, landfill operations, and agricultural mulching (Azeem et al., 2021).

Additionally, wildlife may unintentionally consume plastics not adequately separated before disposal in open landfills within Thai national parks (Teampanpong, 2021), and those plastic items were digested and accumulated in gastrointestinal tract (Teampanpong & Duengkae, 2024), and excreted to feces. Terrestrial vertebrate species may ingest MPs by consuming plants that contain MPs in their stems, leaves, fruits, and flowers from soils (Azeem et al., 2021) and subsequently excrete them in their feces. Therefore, our results align with other studies indicating that wildlife ingests MPs through direct consumption of food, accidental ingestion via contaminated water or sediments (Bessa et al., 2019). Given that most terrestrial vertebrate species in our study are herbivores, it is likely they acquire MPs from plant leaves (Jiao et al., 2024), which may have absorbed MPs from soil (Azeem et al., 2021) or air (Jiao et al., 2024).

Our findings offer preliminary information on potential MPs in terrestrial vertebrate scats in Thailand and Southeast Asia, showing a diverse range of MP abundances in different species. This suggests that terrestrial vertebrate species can excrete both plastics and natural fibers through feces, although data remains limited. For instance, fecal samples from the Sambar deer, a species of conservation concern, showed a range of zero to 15 pieces of potential MPs. Thus, monitoring plastic pollution in feces is an ethically acceptable and non-intrusive method for assessing MP excretion (Thrift et al., 2022) in threatened terrestrial vertebrate species, especially when animal carcasses are rarely found, making reliance on them for MP monitoring impractical.

Study limitations

There are multiple factors that contribute to the limitations of our research. We classify these into two main types: those stemming from the methodologies and those arising from limited research funds. The limitations of research methods can be further classified into field research design, resulting in insufficient sample size for statistical analysis, and methods for quality control and quality assurance in both field and laboratory settings.

The field research design aimed at preliminarily assessing the potential emergence of MPs as a threat to vertebrate species in Thailand resulted in uneven fecal samples and limited MP quantities from fecal samples of various terrestrial vertebrate species across protected and unprotected areas. This led to inadequate sample sizes for statistical analyses on MP occurrence and abundance among species and protected areas.

Identifying potential exogenous MP contamination in both field and laboratory settings was challenging due to the lack of standardized protocols for preparing blanks in quantitative MP research (Shruti & Kutralam-Muniasamy, 2023) at the time of designing this research protocol. We utilized blanks extracted from distilled water, H2O2, atmospheric fallout in the laboratory, and rinsed distilled water from stainless steel spoons in the field. However, we did not collect blanks to test for inherent MPs from transparent PE zip-lock bags and atmospheric fallout during fieldwork. Additionally, we did not conduct MP recovery studies alongside the extraction process or utilize negative controls from the transparent PE zip-lock bags used for field collection, as recommended by Shruti & Kutralam-Muniasamy (2023), Toto et al. (2023), and Way et al. (2022). Furthermore, we neglected to sterilize all glassware and field equipment before use or filter distilled water and all chemicals before the extraction process, as suggested by Álvarez-Méndez et al. (2024). These oversights potentially introduced exogenous MPs into our study.

We found four potential sources of exogenous MP contamination. Firstly, researchers’ attire during fieldwork contributed significantly to MP presence, echoing findings by Scopetani et al. (2020), who reported up to 15% self-contamination from attire in various environments. Secondly, MP deposition emerged as another potential source, supported by our study detecting six airborne microfibers in six out of 10 air samples (0.60 ± 0.52 items per sample), consistent with Donohue et al. (2019), who found a rate of 3.5 ± 2.6 items per sample in air blanks. Unfiltered distilled water used in the extraction process and laboratory cleaning could be a third source, similar to findings by Donohue et al. (2019), who reported MPs in laboratory blanks at a rate of 2.4 ± 2.1 items per sample. Lastly, clear PE ziplock bags used to encase fecal samples in aluminum foil could introduce additional exogenous MPs. Although we identified only two fecal samples with transparent fibers and three fecal samples with fragmented MPs in black, blue, and light blue colors, which were unlikely to have originated from zip-lock bags, the potential contribution remains uncertain. Sterilizing plastic bags before use is recommended to minimize exogenous plastics (Donohue et al., 2019; Katlam et al., 2022). In conclusion, careful handling of fecal samples for analyzing MPs to ensure accurate measurements remain essential even in clean areas (Provencher et al., 2018).

Finally, the limited research fund allowed us to only subsample MPs for verification with FTIR, covering morphologies and colors. This might mean that other suspected items were not identified as MPs. By evaluating only 25.30% of MP samples, it is challenging to accurately identify the predominant polymer types. Therefore, we cannot define the most abundant polymer types of MPs to identify sources of plastics for management. Although we cannot verify if every piece is plastic, other research demonstrates that plastic polymers make up some fractions of the MPs discovered in vertebrates (Hernandez-Milian et al., 2023; Thrift et al., 2022).

Conclusions and future considerations

Western Thailand, where the Thung Yai-Huai Kha Khaeng wildlife sanctuaries are part of the Natural World Heritage Site, is located amidst moderately populated areas and surrounded by rural regions and tourism development. This setting indicates the potential for these sanctuaries to evolve into a biodiversity refuge in the foreseeable future. This study utilized a non-invasive method to characterize, identify, and detect MP pollutants in the feces of terrestrial vertebrate species for the first time in 12 terrestrial wildlife species, including eight that are globally threatened. The potential MPs identified, even in small numbers, were predominantly fibers smaller than 2 mm, primarily black and blue, consisting of both natural and synthetic materials, including rayon, PUR, PET, PP, and fragmented PET and PVC. These MPs likely stem from human-related sources, such as clothing and plastic materials used in agriculture and irrigation systems. We found a correlation between the occurrence of potential MPs in feces and in soil, as well as with human population density.

Given the limited studies on MPs in wildlife, extensive research is needed before firm conclusions can be drawn regarding the significance of potential MPs on terrestrial vertebrate species. Analyzing MPs in feces offers a viable method for gauging plastic pollution in threatened species, allowing for straightforward sampling and identification without relying on animal casualties. This supports ongoing surveillance of plastic pollution in specific wildlife populations. Future research should aim to precisely trace the pathways and mechanisms through which plastics are transferred to these remote areas, impacting wildlife. Monitoring is necessary to understand the effects of these plastic-like substances on terrestrial organisms and the potential impact of plastic additives on animal health. To reduce the spread of microplastic-like substances in Thai protected areas, DNP should encourage the use of natural fibers in clothing and non-plastic materials in facility management. While our results do not conclusively identify plastic pollution as a new threat to biodiversity in these areas, proactive measures combined with public education about the environmental impacts of synthetic fiber usage at the household level are recommended.

Supplemental Information

Supplemental Information 1 Data set for analysis of potential micoplastics in fecal samples of terrestrial wildlife in western Thailand.

(A) is a data set as classified by individual scats. (B) data set using for generalized linear mode (GLM) using negative binomial.

We wish to acknowledge Mr. Kusol Tankjaipitak, the lab technician at the Department of Forest Biology at Faculty of Forestry, Kasetsart University, and Miss Roochira Sukhsangchan, a researcher at the Department of Marine Science, Faculty of Fisheries, Kasetsart University for providing convenient equipment and laboratories and assisting in laboratory operations. We thank Dr. Sampan Tongnunui for supporting some equipment during the field work and help with collecting some fecal samples. We appreciate Waranya Yimprasert, Montita Inja, Darinee Samranruen, Kanyatad Koedpoo, Aiina Rayaphak and Khwanjira Saweswong for field data collection. We thank the hardworking on laboratory operation of Montita Inja (feces), Jiroj Phanchaum, Kaiwad Phanthanu, Aiina Rayaphak and Khwanjira Saweswong (water, sediments, and soils). Special thanks goes to the superintendents of all protected areas where the samples were collected for supporting logistics, accommodation, and suggesting sampling locations. Finally, we appreciate the anonymous reviewers for valuable comments and providing additional research useful for improving our manuscript.

Additional Information and Declarations

Competing Interests

Author Contributions

Animal Ethics

Field Study Permissions

Data Availability

The authors declare that they have no competing interests.

Jiraporn Teampanpong conceived and designed the experiments, performed the experiments, analyzed the data, prepared figures and/or tables, authored or reviewed drafts of the article, and approved the final draft.

Prateep Duengkae conceived and designed the experiments, performed the experiments, authored or reviewed drafts of the article, contribution for laboratory facilities, and approved the final draft.

The following information was supplied relating to ethical approvals (i.e., approving body and any reference numbers):

Institute Animal Care and Use Committee at Kasetsart University provided a full approval for this research (ACKU63-ETC-001).

The following information was supplied relating to field study approvals (i.e., approving body and any reference numbers):

The Department of National Park, Wildlife, and Plant Conservation of Thailand approved the study (ID#0907.4/17863-26-Aug-2020).

The following information was supplied regarding data availability:

The raw measurements are available in the Supplemental File.

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
