# Peer review of "Using feces to indicate plastic pollution in terrestrial vertebrate species in western Thailand"

_PeerJ, doi:10.7717/peerj.17596_

## Round 0.1 · original submission · Major Revisions

The two reviewers have assessed your manuscript and identified a number of issues that make the manuscript unacceptable in its present form. The most important points are that the introduction, results, and discussion sections should be restructured to be more fluent and concise. The results for field blanks and lab blanks should be reported and how these results were considered in the analyses of the samples. Also, the literature does not represent the current knowledge in this field.

Both reviewers underlined the importance of your study, so I hope that their criticisms will allow you to carry out a substantial revision of the manuscript, which is a precondition for acceptance of the manuscript.

**Language Note:** PeerJ staff have identified that the English language needs to be improved. When you prepare your next revision, please either (i) have a colleague who is proficient in English and familiar with the subject matter review your manuscript, or (ii) contact a professional editing service to review your manuscript. PeerJ can provide language editing services - you can contact us at [email protected] for pricing (be sure to provide your manuscript number and title). – PeerJ Staff

Reviewer 1 ·

Basic reporting

The ms provides interesting topics anyway it is ambiguous for some topics not reporting all the results to sustain the conclusions. The intro and Discussion should be improved as well as literature references. Figures are relevant and high-quality but can be improved. In general, the novelty and the aims/hypothesis of the ms should be highlighted. see the comments below, pivotal to improve your ms.

Experimental design

primary research in the scope of the journal. Aims and the sections of ms should be revised. Methods need to be described more in detail and the actual sampling protocol does not allow the Authors to make comparisons for MPs in scats among animal groups. see the comments below (in additional comments), pivotal to improve your ms.

Validity of the findings

The results are interesting and need to be improved. conclusion does not support all results so I would delete some results and add limitations in discussion. see the comments below (Additional comments), pivotal to improve your ms.

Additional comments

General comments:
Here the Authors focused on monitoring MP pollution with scats. I found the paper very interesting with a good flow. I strongly recommend rephrasing some parts to better understand the ms. For instance, the Introduction, results, and discussion should be revised to be more fluent and concise. I strongly suggest dividing the Method, results, and discussion sections considering i. MP in vertebrates scats and ii. Factors determining MP in scats. For results, I would consider deleting some similar parts and divide in two paragraphs as said. Also, I suggest using land use and correlating with the scats found in those precise areas. In general, references in the intro and discussion are outdated and should be more recent and focused. I strongly suggest quoting recent and similar papers on MP in scats for various animal groups.
This paper could increase the knowledge in the field, and before being published needs a MAJOR revision by the Authors.

Specific Comments
I will comment following the Word file lines.
1. Abstract, L22: why “only 40% of the 7 out of 10 vertebrate species”? it sounds weird: does it mean that 3 species were not contaminated by MP? If yes maybe you might rephrase and also explicit better those two concepts giving them more importance – here in abstract and in the ms.
2. L24: how many individuals and how many grams per faeces have your samples? You should provide those info in Methods and results.
3. Abstract, L28: miss a point before “The MP abundance”
4. L30: the gradient of areas comes suddenly: explain before in Methods (in the abstract)
5. L33: The elephant comes suddenly, please provide the readers with a logical flow from the beginning of the abstract
6. L33: here and in results: for this analysis, have you compared the same number of elephant scats and the same number of areas (protected vs non-protected)? Please provide a table in Supp. Mat. to understand that. If you don't have the same number of scats and sites for protected vs non-protected areas you can rephrase the sentences like “MP in elephant scat are more occurring in protected areas than non protected” and also you can discuss it in a paragraph on limitation of your study.
7. L49: here quote more papers like:

- https://www.sciencedirect.com/science/article/pii/S0045653519316303
- https://setac.onlinelibrary.wiley.com/doi/full/10.1002/etc.3432
- https://onlinelibrary.wiley.com/doi/full/10.1002/ece3.9332
8. L51: this is not the right range for MPs, please rephrase it
9. L54: I suggest quoting recent papers:
- https://www.sciencedirect.com/science/article/pii/S0269749121006576
-https://www.sciencedirect.com/science/article/pii/S0025326X23004654
- https://www.sciencedirect.com/science/article/pii/S0048969722078433
10. L62-63: I strongly suggest quoting https://www.mdpi.com/2076-3298/7/10/87 for management
11. L72: Why could work as an indicator of MPs? Please motivate.
12. L73-74: few is known on MP in faeces and terrestrial ecosystems, but quote what has been done. Also in general write that literature on MP in scats is not so abundant.
13. Aims: In general, the novelty and the aims/hypothesis of the ms should be highlighted. Please restructure aims according to results and other ms sections.
14. L95: check typo: “Cosnervation”, check typo also along the ms
15. L129: Please specify from where have you adapted this protocol from the literature. Because there are many published protocols. On this topic please consider:
- https://doi.org/10.1016/j.scitotenv.2018.07.101
- https://link.springer.com/article/10.1007/s11356-022-21032-0
16. L130: there were any plastics on the scats? You should have kept it off the scats because of external contamination..
17. L131: what about the weight of the total scats sampled? And per animal group? You can report it in Supp Matt.
18. L141: why sampling other matrices? What are the aims? If you need it, please consider adding to your aims.
19. L163: why have you selected MP for FTIR by types and colors? I would rely on where those potential MP were found – so considering the abundance in each animal group.
20. Results: where are data on MP in matrices (water, sediment, etc)? If not present, please eliminate this analysis from methods and discussion. I would focus only on scats.
21. L207-214: please rephrase here and everywhere in the ms: you should revise the concept of “the highest number of MPs is in this scat animal than this other one” because you have not collected the same number of scats per animal group. Please add it to a paragraph of limits of your study (“Given that we have not collected the same number of scats per animal group, we cannot have a proper and valuable standardization and so we cannot compare plastic concentration among animal groups..etc etc”). Thus, this part of the results should be rephrased, and you can add a table or figure with the number of plastic concentrations per animal group, but without comparing.
22. L215: all the subheadings in results are not expected, you should introduce them from methods so that readers can better follow your findings. I suggest splitting not too many methods, results and discussion (for instance there is no need to have a specific subheading on the elephant, and maybe you can merge the MP characteristic in one subheading/paragraph on colors, shape, ect. Also, do not repeat information for each MP characteristics and each animal group but provide a table/figure to synthesise all. I strongly suggest dividing the Method, results, and discussion sections considering i. MP in vertebrates scats and ii. Factors determining MP in scats.
23. L299: here “there 77 MPs recorded” and in other parts please rephrase.
24. L301: to these 7 species, right? As only 7 species scats were contaminated by MPs..
25. In general, references are outdated and should be more recent and focused. I strongly suggest quoting recent and similar papers on MP in scats for various animal groups, such as:
- https://doi.org/10.1016/j.scitotenv.2018.07.101
- https://link.springer.com/article/10.1007/s11356-022-21032-0
- https://www.sciencedirect.com/science/article/pii/S0048969721014637
26. Some papers to enhance the discussion:
- https://www.sciencedirect.com/science/article/pii/S0048969722037767?casa_token=Yc0lTBj72jwAAAAA:WCaV0_pgDZffjRPaoZiGcY8GiFCUngmlA65sqUOY00oIWIj1NTm3uWg4JQeUwX1uIBz3OLxbXA
- https://doi.org/10.1016/j.marpolbul.2023.115227
27. L381: please add in limitations that data are scanty anyway these are preliminary information on MP in vertebrate scats in Thailand.
28. Conclusion: concerning all these analyses/findings, please consider which gives more focus meanwhile restructuring and revising your ms.
29. Fig.3: please add “A,B,C” in the caption rather than upper, middle, or lower. Also, I strongly suggest adding the mean values with histogram (so MPs/ind or MPs/scat gram) instead of the number of MPs.
30. Fig4: check the caption, there is a typo (charactericistics).
31. Figure 5 and in the result section: please consider using and correlating MP found in scats with the land use (near where you collected the scats). I suggest adding this of land use as you have MPs in scats found in remote areas, so it will be interesting to highlight this result, and also to discuss MP in relation to remote areas (and land use).
32. Fig.5: why for elephants? I would do for all animals of for the top-3 most occurring? Please motivate it.
33. Discussion: please restructure your discussion as i. MP in vertebrates scats and ii. Factors determining MP in scats. Also, discuss with more literature and recent papers.
34. Supp.Matt. (and in results): if you have found plastics with size > 5 mm you should emphasise that you found macroplastics in faeces, and please put also a picture of that in supp. Matt.
35. At the end, in the acknowledgement, add the role of anonymous reviewers to improve the draft of your manuscript.
36. a check of the english flow should be done, thanks. ask to an external colleague eventually.

Reviewer 2 ·

Basic reporting

Accumulation in the toxicological sense is when something builds up, and is not metabolised or excreted. This paper is about excretion of MPs via the feces. Suggest changing the wording in the paper to reflect this terminology of the field.
The analyses of the MPs in elephant feces in protected and not protected areas is interesting – but this shows that the MPs ingestion is likely widespread, which is what we expect based on what has been found in other mammal studies. I would suggest adding this as an objective in the intro, and making a prediction. Why was this tested, you must have expected a difference?
While the authors explain the lack of information on MPs in biota in the terrestrial systems, there are several papers that have explored MPs in terrestrial animals that could be drawn on.
There are also recent papers on MPs in mammals that could be more useful to cite that while these animals are big, and likely exposed, they don’t actually accumulate a lot, and their fecal levels are relatively low.
Understanding the occurrence and fate of microplastics in coastal Arctic ecosystems: The case of surface waters, sediments and walrus (Odobenus rosmarus) - PubMed (nih.gov)

Experimental design

The authors needs to explain more both field blanks and lab blanks were treated and considered within this study. How many were used? What exactly did the blanks look like? It seems like the blanks were the material used, but don’t account for the process or the field? Was the data blank corrected before or after spectroscopy? Without these controls, the reader can’t have confidence in the reporting of microfibers, which are well known to happen due to contamination of the samples during the collection and processing.
Sustainability | Free Full-Text | Cross-Contamination as a Problem in Collection and Analysis of Environmental Samples Containing Microplastics—A Review (mdpi.com)

Validity of the findings

Generally, until the issue of the blanks is considered more completely, I am not sure how to interpret the results, and therefore didn’t really read the discussion on depth. Many excellent points are made, but unless we are confident in the data, it is hard to address.
If no field blanks were collected, then the authors should limit their results and discussion to those pieces that are not fibers. Those morphologies can be confidently reported without blanks. Given the low values of the numbers found in the feces, for at least some of these animals, I suspect that they are not above the limit of detection, which are now being recommended to be considered in more depth for environmental sampling.
How to establish detection limits for environmental microplastics analysis - PubMed (nih.gov)

Additional comments

Until this work addresses the blank consideration in more detail, publication should not proceed.

---

## Round 0.2 · Minor Revisions

Although the quality of the manuscript is now improved, there are still some points that have to be addressed before an acceptance of your paper, as outlined in the two reviews. Please follow the reviewers' suggestions or address the reasons why you do not follow in your rebuttal letter.

Reviewer 1 ·

Basic reporting

I strongly suggest implementing the State of the Art and the discussion of the findings. Moreover, clarifying methods will be pivotal to understanding their findings. A better visualization of results (i.e., figures) is important to reach more easily readers and the scientific community – I request a large improvement of their figures to provide the ms high-quality. FTIR spectra of dominant polymers should be added to the ms.

see report below

Experimental design

see comments below

Validity of the findings

A better visualization of results (i.e., figures) is important to reach more easily readers and the scientific community. FTIR spectra of dominant polymers should be added to the ms.

Additional comments

To The Authors:
The Authors answered most of my comments. Thanks to the Author’s efforts, the ms quality has been largely improved. However, I strongly suggest implementing the State of the Art and the discussion of the findings. Moreover, clarifying methods will be pivotal to understanding their findings. A better visualization of results (i.e., figures) is important to reach more easily readers and the scientific community – I request a large improvement of their figures to provide the ms high-quality. FTIR spectra of dominant polymers should be added to the ms. For the details, follow the below comments.
After those suggestions are made, the paper could be considered for publication.
Have a fruitful job.

Detailed comments:
1. Authorship: does it is possible to have both first and last author?
2. Title and abstract: I would say that you focused on “terrestrial vertebrate species” . Check that the word lenght in the abstract is ok.
3. I noticed that you added results by FTIR analysis: I strongly suggest adding a Figure in Supp. Mat. Of the instruments you used and also a figure of spectra of most occurring polymers.
4. Your introduction missed some references:
so, L96 needs a reference
5. L98: pay attention on how to quote references. For instance “Weinstein, Crocker & Gray, 2016” is Weinstein et al. 2016, so correct here and along the ms.
6. Please check your hypothesis and motivate as hypothesis. Probably I would emphasise that I would expect more plastics in scats recollected from not natural/urban areas rather than pristine areas, right?
7. L183: if you choose this order (vertebrate scats, water, soils, and sediment,) please maintain it in all the ms sections (methods, restults, discuss) and also in the figures
8. L189: what do you mean for opportunistic sampling? Please rephrase this.
9. Methods, L200: have you collected the same grams of scats for all the species? Have you standardised on the same number of scats recollected? If not, report it in methods and specify in discussion that your results are based on different grams of scats per species (and number of scats), so your resutls are not comparable among species, but it’s a mapping of MPs in scat’s vertebrates in Thailand.
10. Please report all the references used for field collection of MPs in the different matrices. The same should be done for the protocols for lab processing of those matrices to extract MPs.
11. L229: thawed? Please rephrase it
12. L290: how is possible that you analise the water of fieldwork if you did it many years ago? Please specify more or rephrase. Then have you substracte the MPs found in blanks from the real MPs of your findings?
13. Data analysis: some references when you performed GLM? Please, also consider correcting your analyses and results with the different number of species in the GLM test.
14. Data analysis: all these elements (proximities to landfills, tourist sites, local landmarks, villages, and unpaved roads.) come suddenly, so you have to specify them in intro and in your hypothesis’s aim.
15. Results: I strongly recommend shortening the results: I mean that all the number of MPs concentrations can just be kept in the Table and indicate in the text some general patterns (referring to the Tables).
16. Results: You must add a figure with the spectra of the most dominant polymers verified at FTIR and found in your samples. This figure can be added in Supp. Mat. If better. Moreover, in the results, add the % of dominant polymers found in your samples.
17. In discussion: please add a limitation paragraph to discuss the limitations of your study. For instance, the low number of MP found to perform statistics and provide with data-driven insights. And also that you’re using a subsample of MPs verified at the ftir, and that the other suspected MPs we don’t know what they are.
18. Figures: fig.2, you can consider inserting the image of animals in the x-axis (the same image you used for animals in Fig.1) – for instance you can follow Fig. 4 by this study:
https://link.springer.com/article/10.1007/s11356-023-26617-x
Fig. 3 is too complex to be immediately get. Therefore, I strongly suggest splitting the panels and thinking on putting a Figure in Supp. Mat. Or otherwise to make a unique Figure for plastic size but using histograms
Fig.4, please indicate where are the macroplastics
Fig. 5, what about the colour? I would strongly suggest putting it near the stattions of natural areas and then the others by more urban areas. If you used acronym in study area for the stations, please use also here.
Fig. 6: consider that is not too easy to digest as the high quantities of data. I would suggest putting here the images of animals to make the figure more catchy and easy to be got by readers.

Reviewer 2 ·

Basic reporting

I don't think that items.ind-1 is the right metric, I think that you want to report on a per gram basis of scat, or on a per scat basis. items.ind-1 implies that the individual excreted their plastic burden, but this is not likely.

The authors state that they expected high levels of black pieces, but can you explain why?

Line 343 - need to check, I don't believe that the dominant plastic type in fulmars is fibers, I think it is reported to be fragments as they are relatively large birds.

line 433 - need a space before Analysing

Experimental design

I am still not clear on the field blanks that are used. The paper states that "Additionally, 5 field blanks of distilled water used for cleaning metal spoons and soil cores were tested". But I don't understand this. Field blanks should be just blank empty containers similar to what you put your samples in. They are treated the same way as the sample containers, but they don't get samples. This tells you if your field collections contributed to the MPs in your samples.

What was the type of plastic bag used? If you know the polymer, but don't have field blanks, you could just remove any reports of this polymer.

More clarity is needed on the air blanks. Why were the blanks collected over a longer period? Are these air samples in the lab? I don't understand how this data is relevant here. Or are these the procedural blanks?

Validity of the findings

I don't see the water and soil results anyway. Given that you include methods for these, I suggest also adding in the results.

I think that the fact that the authors found so many types of polymers also helps align with the fact that the results are not from cross-contamination. Typically it comes from the bags used, and this would all be one type of bag, and thus you get a really dominant plastic type. I think this could be emphasized more.

---

## Round 0.3 · accepted · Accept

Thank you for the revision of the manuscript. I hereby certify that you have adequately taken into account the reviewers' comments and improved the manuscript accordingly. Based on my assessment as an Academic Editor, your manuscript is now ready for publication.